# We're all in this together: Focus on community attenuates effects of pandemic-related financial hardship on reactance to COVID-19 public health regulations

Michael E. Knapp[1]*, Lindsey C. Partington[2], Ryan T. Hodge[2], Elisa Ugarte[2], Paul D. Hastings[2]

1 Division of Behavioral and Organizational Sciences, Claremont Graduate University, Claremont, California, United States of America, 2 Department of Psychology, University of California, Davis, California, United States of America

☯ These authors contributed equally to this work.
‡ These authors also contributed equally to this work
* michael.knapp@cgu.edu

**Data Availability Statement:** The data and full reproducible code can be accessed via https://github.com/meknapp18/ProReact.

## Abstract

There has been resistance to COVID-19 public health restrictions partly due to changes and reductions in work, resulting in financial stress. Psychological reactance theory posits that such restrictions to personal freedoms result in anger, defiance, and motivation to restore freedom. In an online study (N = 301), we manipulated the target of COVID-19 restrictions as impacting self or community. We hypothesized that (a) greater pandemic-related financial stress would predict greater reactance, (b) the self-focused restriction condition would elicit greater reactance than the community-focused restriction condition, (c) reactance would be greatest for financially-stressed individuals in the self-focused condition, and (d) greater reactance would predict lower adherence to social distancing guidelines. Independent of political orientation and sense of community, greater financial stress predicted greater reactance only in the self-focused condition; the community-focused condition attenuated this association. Additionally, greater reactance was associated with lower social distancing behavior. These findings suggest that economic hardship exacerbates negative responses to continued personal freedom loss. Community-focused COVID-19 health messaging may be better received during continued pandemic conditions.

## Introduction

Government-mandated or encouraged public health regulations to decrease the community transmission of the COVID-19 virus have elicited diverse reactions from the populace of the U.S. The effectiveness of such regulations for decreasing transmission is contingent upon the acceptance and cooperation of the population. Whereas many have abided, others have refused these regulations or guidelines as illegitimate and inappropriate government overreach

**Funding:** The author(s) received no specific funding for this work.

**Competing interests:** The authors have declared that no competing interests exist.

infringing upon their rights [1, 2]. Why, and what can be done to reduce reactance against public health measures in the face of a global pandemic? Psychological reactance theory (PRT) posits that restriction or loss of behavioral freedoms results in anger, negative cognitions, and motivation to restore freedoms [3–5]. We propose that adults who have experienced greater economic hardship due to COVID-19 public health regulations will evince greater reactance, but that focusing their perspective on the regulations' impact on their community–rather than personal impact–will attenuate that reactance.

## Reactance in the pandemic context

PRT proposes that maintaining agency and autonomy, known as *behavioral freedom*, is a universal human motivation [6]. Restricting or eliminating behavioral freedom generates *reactance* [3, 4], both when freedoms are explicitly restricted (e.g., state-enforced curfew) as well as when restrictions are implied (e.g., messages to limit one's own behavior; [3, 7]. Public health campaigns to reduce unhealthy behavior and to promote health behavior have been found to generate reactance [8–11]. This may be due to messaging being seen as threatening to personal freedom for health behaviors, as the magnitude of reactance experienced is associated with the freedom's perceived importance and extent of threat [5]. As such, PRT presents a potential explanation for resistant or defiant responses in the COVID-19 pandemic context. Public health regulations that were experienced as restricting or removing highly-valued personal freedoms would have been experienced as threatening, and therefore likely to engender greater reactance. For example, regulations such as social distancing, limited capacity in stores and workplaces, and designations of (non-)essential work status may directly impact individuals' abilities to meet routine life demands such as paying rent or mortgage, buying food and essential supplies, and supporting other financial needs [12]. The effects may be particularly damaging when the restrictions to behavioral freedom have the potential to be long-lasting.

According to PRT, reactance may prompt behavioral resistance wherein individuals attempt to restore and maintain personal autonomy and behavioral freedom by continuing to engage in the restricted behavior, endorsing negative attitudes or opinions about the imposed restrictions, or actively arguing against or seeking out information that refutes the restrictions (for review, see Rosenberg & Siegel, 2018). Persuasive messaging aimed at eliciting behavioral change may also instigate reactance and associated behavioral resistance [13] resulting in a "boomerang effect" wherein message recipients adopt the opposite behavior [14]. In the context of COVID-19, this "boomerang effect" has been reflected in protests against and outright refusal to follow public health guidelines [15, 16]. Burgeoning research suggests that reactance is associated with decreased adherence with social distancing, hand-washing, and mask-wearing [17]. Moreover, reactance to COVID-19 public health measures has been associated with negative attitudes towards masks, increased beliefs about masks being ineffective, and outright refusal to wear a mask [18]. Consequently, utilizing messaging that reduces reactance to public health mandates is a potential mechanism for increasing compliance with these preventive health measures.

## Avenues for attenuating reactance: Other-oriented messaging

Reactance is reduced when people are directed to focus on how others, rather than oneself, are affected by restrictions to freedom, known as *vicarious reactance* [19, 20]. Self-reactance may activate the affective system, leading to impulsive responses, whereas vicarious reactance activates cognitive systems, enabling reflection upon another's plight with less emotional arousal [21]. Indeed, health messaging that induces empathy or other-oriented thinking is often more persuasive and more successful in eliciting behavioral change as it curbs reactance [14, 22].

Extending this to the current pandemic, public health messaging that focuses attention on how restrictions impact one's community, rather than oneself, may spur vicarious reactance rather than personal reactance, thereby reducing anger, impulsive actions, and motivation to re-establish freedoms, even among individuals who have been adversely impacted by these restrictions. Consistent with this hypothesis, current public health messaging has adopted a community-focused approach [23], but the extent to which this technique is effective in reducing reactance during the pandemic has yielded conflicting findings.

Specifically, recent research has shown that prosocial messaging with a focus on community, compared to focusing on the self, has had mixed effects on COVID-19 prevention behavior with U.S. samples (for review, see [23]). Some studies have found that messaging that focused on the community increased intentions to wear a mask [24] as well as increased prevention intentions (e.g., handwashing; [25]). However, findings have not been consistent across health behaviors, as message framing that focused on community did not have similar effects for social distancing and understanding COVID-19 through official communication [23]. Similarly, neither Favero and Pedersen [26] nor Ma and Miller [27] found significant differences between self-focused and empathy-inducing or other-focused public health messages on social distancing intentions and other COVID-19 attitudes and beliefs.

One possible explanation for these inconsistent findings is that other factors may influence, or moderate, the impact of public health messaging [27]. That is, contextual factors specific to or resulting from the COVID-19 pandemic and the associated changes to daily life may influence how receptive an individual is towards the framing of pandemic public health messaging. Indeed, Luttrell and Petty [28] found that other-focused public health messages were rated as more persuasive than self-focused messages but only for participants who strongly moralized public health. The present study investigated whether individuals' experiences of acute, pandemic-related financial stress influenced how the framing of regulations as affecting the community or the self predicted reactance to ongoing and future regulations.

## Pandemic-specific financial stress as a moderating influence

COVID-19 not only introduced a national health threat but an economic threat as well [29, 30]. With business closures and workhour reductions, 70% of U.S. families experienced detrimental work impacts by April 2020 [31] leading to greater financial stress [32] and decreased ability to afford basic needs [12]. Thus, public health regulations that limited work activities may have been perceived as personally threatening. Media coverage of hoarding goods and supply chain disruptions may have exacerbated perceptions of threat and need to prioritize self-interests, especially for those experiencing greater financial stress [33]. In addition to the threats to physical health posed by the COVID-19 virus itself, the economic repercussions of the pandemic challenged many people's financial security.

It is possible that individuals experiencing greater pandemic-specific economic hardship and financial stress may be particularly sensitive and reactive towards messages about pandemic preventative measures, especially those that may infringe upon their livelihood (e.g., non-essential business closure, shelter-in-place). Accordingly, public health regulations that were experienced as both limiting personal agency and jeopardizing financial stability would have been experienced as particularly threatening, and therefore likely to engender greater reactance followed by decreased public health compliance in an effort to reclaim behavioral autonomy. Conversely, public health messaging that orients financially stressed individuals to focus on their community may attenuate reactance as the individual reflects upon pandemic impacts on others in the community.

## Critical covariates: Individual differences in pandemic response

Whereas we expected that COVID-related financial stress could moderate the extent to which self- versus community-focused messaging affects reactance, other individual differences could be expected to affect people's likelihood of reactance to public health measures, regardless of messaging. Three such factors were considered in this study: psychological sense of community (PSOC), political orientation, and gender.

Many people have complied and cooperated with government-mandated or encouraged public health regulations. Those following regulations may feel greater PSOC, or connection to groups impacted by COVID-19 [34, 35]. Greater PSOC is associated with engaging in behaviors that benefit the community while avoiding those that harm [36]. PSOC with others affected by COVID-19 –be they frontline workers, families of infected people, or others–may be expected to curb reactance to loss of freedom through government mandates. Previous research has suggested that cooperation during times of crisis is motivated by shared identity and concern for others [37]. Therefore, PSOC was included as a covariate in this investigation.

In addition to what may facilitate pandemic prevention behaviors, contextual features in the current political climate regarding the emergence, presence, and threats of COVID-19 in the U.S. should be considered. During 2020, in the hyperpolarized political context of the U.S., public health measures were not seen as values-neutral, but as reflective of political orientation [38]. This phenomenon may have been amplified by the ease with which people can select the source of their news, filtering out discrepant messages in favor those confirming one's position, and thereby adopting or increasing partisan views regarding COVID-19, public health measures and their own health behavior choices [37]. Research on climate change messages suggests that political orientation may be associated with reactance towards public policies based on scientific consensus ([39]; but see [40]). Political orientation may make some individuals less cooperative with public health messaging that contradicts their beliefs about COVID-19, or the actions they see encouraged by the political agents with which they identify. Within the U.S., research on political orientation and pandemic-related attitudes and behaviors suggests Republicans and more politically conservative people perceive less health risk due to COVID-19, whereas Democrats and more politically liberal people are more compliant with public health directives [41, 42]. Consequently, we incorporated political orientation as a second covariate in this investigation.

Finally, recent research into COVID-19 has highlighted gender differences in public health compliance. In general, men tend to engage in fewer preventive behaviors to protect health [25] and more health risk behaviors [43] as compared to women, which may contribute to decreased COVID-19 public health compliance in men. In early June 2020, when mask wearing was recommended by public health officials but prior to widespread mask mandates, women had increased odds (1.5x) of voluntarily wearing masks to a store as compared to men [2]. Similarly, Capraro and Barcelo [24] found that men had less intention to voluntarily wear a face mask and more negative feelings about wearing a mask as compared to women. Recent meta-analytic evidence suggests that rates of mask wearing have been comparable for men and women, but men are more likely to perceive masks as an infringement on their personal freedom [44]; hence, men may have increased reactance toward COVID-19 public health mandates and recommendations. Therefore, we included gender as an additional covariate in our analyses.

## Current study

We examined whether self- versus community-focus of restrictive COVID-19 public health messaging affected reactance, and how pandemic-related financial stress was associated with

reactance against public health regulations. Additionally, we examined the extent to which reactance towards COVID-19 public health restrictions was associated with social distancing behavior. These effects were explored using an online study between May 7 and June 15, 2020, after widespread U.S. shelter-in-place orders had been established and maintained for several weeks, while controlling for pre-pandemic income and individual differences shown to be associated with pandemic response–specifically, PSOC, political orientation, and gender. We hypothesized that individuals reading self-focused messages about public health restrictions would report greater reactance compared to individuals reading community-focused messages (H1), that pandemic-related financial stress would predict greater reactance (H2), and that the association between financial stress and reactance would be reduced for those in the community-focused condition compared to those in the self-focused condition (H3). Finally, we hypothesized that greater reactance would be associated with less adherence to social distancing guidelines (H4).

## Method

### Participants

Recruitment was based upon convenience sampling of adults (18 years or older) with access to a computer with internet capabilities. From April 30 to June 15, 2020, advertisements for "an online study to better understand people's emotional and behavioral responses to the COVID-19 pandemic and the associated public health regulations" were placed on Internet networking sites such as Facebook and disseminated through UC Davis venues. A power analysis to determine an adequate sample size was determined using G\*Power for multiple linear regression ($R^2$ increase) testing three predictors in a six predictor model. Parameters were set at an alpha of .05, effect size of .10, and power of .90, which resulted in a minimum of 146 participants to achieve this effect. A total of 386 adults consented to participate (Age; $M$ = 35.74, $SD$ = 14.78, *Range*: 18–87, 76.04% female, 69.71% White); however, a subset of 301 adults were used in the subsequent analyses (Age; $M$ = 35.88, $SD$ = 14.34, *Range*: 18–87; see data cleaning for detailed subsetting process). The sample used for subsequent analyses primarily identified as White ($n$ = 218, 72.91%) and female ($n$ = 232, 77.08%) from 17 states and Washington, D.C. (77% California, 4% Colorado, less than 4% for any other state, 4% declined to report). The sample used for subsequent analyses was diverse in their reported income levels, ranging from less than $20,000 to more than $175,000 with each category containing at least 18 participants. However, the largest grouping was between $20,000 - $80,000 (three intervals; $n$ = 125, 41.52%).

### Procedure

The UC Davis IRB Administration approved the following project under IRB ID 1587525–1 and determined the project as Exempt 2 status. Consent was obtained from all participants prior to participation. Participants completed the study using Qualtrics, an online data collection tool. Public access to the survey began one week after the first online advertisements, with data collected from May 7, 2020 through June 15, 2020. Prior to completing the survey, participants viewed an online informed consent page. After providing consent, participants completed questionnaires in the following order: demographics (including gender and income), financial stress, adherence to social distancing, political orientation, and psychological sense of community. Participants completed an attention check midway through the survey. Next, Qualtrics randomly assigned participants to read one of two restriction of freedom scenarios, self-focused restriction or community-focused restriction. After reading the scenario, participants answered questions regarding their reactions and responses to what they just read.

Participants were thanked for their time and participation at the end of the survey. There was no financial incentive for participation. The survey took participants approximately 15–20 minutes to complete.

## Measures

**Restriction of freedom scenarios.** Participants were randomly assigned to one of two conditions: self-focused restriction or community-focused restriction. Each condition involved reading a prompt describing COVID-19 public health regulations (e.g., social distancing, closure of non-essential businesses) and asking the participant to imagine how these restrictions will continue to impact freedom. In the self-focused condition, participants read a prompt describing how the COVID-19 public health regulations impacted themselves and asking the participants to imagine how these restrictions would continue to impact their own freedom to work, have a social life, and go about daily life (Fig 1A). In the community-focused condition, participants read a prompt describing how the COVID-19 public health regulations impacted others in the surrounding community and asking the participants to imagine how these restrictions would continue to impact community members' freedom to work, have a social life, and go about daily life (Fig 1B).

**State reactance.** We adapted the Salzburger State Reactance scale [45] to measure participants' reactance to the restriction of freedom scenarios. The scale consisted of 11 items assessing experience of reactance (e.g., *"To what extent do you perceive these actions of the government as a restriction of freedom?"*), aggressive behavioral intentions (e.g., *"Do you think people should be ignoring, resisting, or protesting these restrictions, limitations, and guidelines?"*), and negative attitudes (e.g., *"How likely do you think it is that the government will use these actions to take advantage of people?"*). Participants answered questions using a 5-point Likert-type scale (1 = *Not at all*, 5 = *Very Much*). Mean scores for all items were used to index

**A. Self-Focused Restriction Condition**

> *Please consider how the recent developments in the spread of COVID-19 have impacted your daily life and your future. Among these developments have been restrictions that the government has placed on you directly which includes restricting of your physical space with others (social distancing), cleanliness with washing your hands and avoiding touching your face, and limiting you to visiting only essential locations.*
>
> *Take a moment to imagine how these restrictions will continue to impact you in the future. The government is likely to continue to restrict your behaviors. It is certain that these restrictions will continue to greatly impact your freedom to work, have a social life, and go about your daily life.*

**B. Community-Focused Restriction Condition**

> *Please consider how the recent developments in the spread of COVID-19 have impacted the daily lives and futures of the community around you. Among these developments have been restrictions that the government has placed on others around you directly, which includes restrictions of their physical space with others (social distancing), cleanliness with washing their hands and avoiding touching their face, and limiting them to visiting only essential locations.*
>
> *Take a moment to imagine how these restrictions will continue to impact the community in the future. The government is likely to continue to restrict their behaviors. It is certain that these restrictions will continue to greatly impact their freedom to work, have a social life, and go about their daily life.*

**Fig 1. Vignettes describing the impact of COVID-19 public health restrictions.** Participants were randomly assigned to be in either the self-focused restriction condition (A) or in the community-focused restriction condition (B).

participants' state reactance to the restriction of freedom scenarios. Cronbach's alpha = .90 with bootstrapped 95% CI [.87, .92] based on 1,000 iterations.

**Adherence to social distancing.** Participants' adherence to public health guidelines for social distancing was measured with their responses to the item, "Since the pandemic began, I have tried my best to socially distance from others to decrease viral infection in the community". Participants were asked to rate the extent to which they agreed with the statement using a 7-point Likert-type scale (1 = *Strongly Disagree*, 7 = *Strongly Agree*). This item was created by the researchers for the purposes of the study.

**Financial stress.** Participants' financial stress was measured using 6 items derived from Conger and colleagues' economic hardship measures [46, 47]. The prompt "Since the pandemic began" was added to items to orient participants to rating financial stress experienced in the wake of the pandemic. The measure included two items assessing inability to make ends meet (e.g., *"Since the pandemic began, you have been late paying your bills or have not been able to pay your bills"*), two items assessing unmet material needs (e.g., *"Since the pandemic began, you have not had enough money to afford the kind of food you needed"*), and two items assessing financial strain (e.g., *"Your financial situation is worse now than it was before the pandemic began"*). Participants were asked to rate the extent to which each statement was true for their circumstances using a 5-point Likert-type scale (1 = *Never; not at all true*, 5 = *Almost every day, definitely true*). Mean scores represented participants' overall financial stress. Cronbach's alpha = .85 with bootstrapped 95% CI [.79, .88] based on 1,000 iterations.

**Gender.** Participants' gender was measured as their response to the following question, *"What gender do you identify with?"*. Response options included "female", "male", "nonbinary or genderqueer", "transgender", "other–please specify", and "decline to answer". Of the 301 subsetted participants used for analyses, four identified as nonbinary or genderqueer, one identified as other, and one declined to answer. Given the unequal numbers in gender identities, we re-coded responses into a binary variable with 1 indicating female and 0 indicating not female. The one participant who declined to answer was marked as missing data.

**Political orientation.** Participants' political orientation was assessed with their responses to the question, "In general, how would you describe your overall political orientation?". Using a 7-point Likert-type scale, participants were asked to indicate the response that best described them (1 = *Very Liberal*, 4 = *Moderate*, 7 = *Very Conservative*). Higher scores represented more conservative political leaning.

**Psychological sense of community.** We adapted four items from a measure of psychological sense of community measuring connection and identification [36] to assess participants' psychological sense of community with those affected by the pandemic (PSOC). Two items measured participants' connection to those impacted by COVID-19 (e.g., *"I feel strong ties to the community affected by COVID-19"*) and two additional items measured participants' personal identification with those impacted by COVID-19 (e.g., *"I identify with the community affected by COVID-19"*). Participants' indicated the extent to which they agreed with each statement using a 5-point Likert-type scale (1 = *Completely Disagree*; 5 = *Completely Agree*). Mean scores for all four items represented participants' overall COVID-19 PSOC. Cronbach's alpha = .93 with bootstrapped 95% CI [.91, .95] based on 1,000 iterations.

**Income.** Gross household income in 2019 was measured using 9 intervals starting at "Less than $20,000", "$20,000 to $40,000", and so on up to "More than $175,000." Participants were additionally given an option "Don't know/Decline to answer." Given the primary focus on financial stress, income was included as another covariate in the tests of hypotheses.

### Analytic plan

All analyses were performed in R 1.3.1093. Preliminary analyses were performed with the *psych* [48] and *jmv* packages [49]. Path analyses were performed with the *lavaan* package [50] with simple slopes performed with the *semTools* package [51].

**Data cleaning.** The data cleaning procedure included handling missing data as well as removal of participants who did not complete the restriction manipulation, reactance measure, or those who failed the attention check. Of those who consented to participate, 45 discontinued before receiving the restriction manipulation, 7 discontinued without completing the reactance measure, and an additional 5 failed the attention check, which reduced the sample to 329 participants. Finally, 28 participants were removed on the income measure for not knowing or declining to report their income (i.e., selected "don't know/decline to answer" as their response), which resulted in a final data set for analyses of 301 adults. On the variables of interest, Little's MCAR test was conducted, $\chi^2(5) = 11.06$, $p = .05$, and confirmed the data was missing completely at random. Of note, the only variable with missing data was financial stress (percentage missing 1.96%). Thus, given the small amount of missing data, full information maximum likelihood (FIML) was used during path analysis to account for missingness.

**Assumptions.** The analysis plan included using path analysis with a binary-continuous interaction term between financial stress and restriction condition. The multivariate normality assumption for path analysis regression was not met as demonstrated by the significant Mardia's skewness ($p < .001$) and kurtosis ($p < .001$; [52]). Given the violation of multivariate normality, we used Maximum Likelihood with robust corrections to fit the model and obtain parameter estimates with robust standard errors [53, 54].

### Open practices statement

The current analysis was not formally preregistered. The data and full reproducible code can be accessed via https://github.com/meknapp18/ProReact.

## Results

### Descriptive statistics

Table 1 contains zero-order correlations by restriction condition for all key study variables, as well as means, standard deviations, skew, and kurtosis. For the entire sample, state reactance was positively correlated with financial stress ($r = .19$, $p = .001$) and political orientation ($r = .36$, $p = .034$) and negatively correlated with PSOC ($r = -.28$, $p < .001$), income ($r = -.12$, $p = .04$), and adherence to social distancing ($r = -0.42$, $p < .001$). State reactance was not significantly correlated with gender ($r = .02$, $p = .71$). For the self-focused condition ($n = 148$), state reactance was positively correlated with financial stress and political orientation, was negatively correlated with PSOC and with adherence to social distancing, and was not associated with income or with gender. For the community-focused condition ($n = 153$), state reactance was positively correlated with political orientation, negatively correlated with PSOC and with adherence to social distancing, and not associated with either financial stress, income, or gender.

As a check on the randomization process, we performed planned comparisons with between-subjects' *t*-tests to probe for differences between restriction conditions on financial stress, PSOC, political orientation, income, gender, and adherence to social distancing. Those in the community-focused condition were more politically conservative ($M = 2.99$, $SD = 1.46$) compared to those in the self-focused condition, $M = 2.57$, $SD = 1.53$, $t(299) = -2.44$, $p = .015$,

**Table 1. Descriptive statistics and zero-order correlations for reactance and predictors.**

| Variables | 1 | 2 | 3 | 4 | 5 | 6 | 7 |
|---|---|---|---|---|---|---|---|
| 1. Reactance | — | −.36*** | .30*** | .03 | .37*** | −.29*** | −.13 |
| 2. Adherence to Social Distancing | −.50*** | — | −.21** | −.01 | .23** | .26** | .18* |
| 3. Financial Stress | .08 | −.22** | — | .02 | .02 | .01 | −.43*** |
| 4. Gender | −.07 | .18* | .04 | — | −.01 | −.12 | .08 |
| 5. Political Orientation | .34*** | −.31*** | .07 | −.01 | — | −.09 | .15$^t$ |
| 6. PSOC | −.29*** | .11 | −.10 | −.01 | −.01 | — | −.01 |
| 7. Income | −.11 | .13 | −.23*** | .07 | .04 | .16* | — |
| M | 2.32 | 6.30 | 1.48 | 0.80 | 2.78 | 3.26 | 4.91 |
| SD | 0.80 | 1.07 | 0.72 | 0.40 | 1.51 | 1.01 | 2.60 |
| Skew | 1.15 | −2.28 | 2.17 | −1.48 | 0.83 | −0.53 | 0.23 |
| Kurtosis | 1.32 | 6.40 | 4.65 | 0.20 | 0.14 | −0.17 | −1.16 |

$N$ = 301. Zero-order correlations for the self-focused condition ($n$ = 148) are reported above the diagonal. Zero-order correlations for the community-focused condition ($n$ = 153) are reported below the diagonal. PSOC = Psychological sense of community. Gender coded 1 = Female, 0 = Not Female.

$^t$ $p < .10$,

$^*$ $p < .05$,

$^{**}$ $p < .01$,

$^{***}$ $p < .001$.

95% CI: [-0.76, -0.08]. There were no other significant differences between restriction conditions on predictors and covariates ($p$'s > .05).

## Path analysis

A path model included direct pathways from all exogenous study variables (e.g., financial stress, dummy-coded restriction condition, etc.) and the interaction of financial stress and restriction condition to state reactance. In addition, based on the correlation matrix, covariance between restriction condition and political orientation, financial stress and income, and the interaction term and its components were specified. Both standardized (Table 2) and unstandardized path coefficients (Fig 2) were estimated, resulting in model fit indices of: $\chi^2(24) = 44.03$, $p = .01$, robust CFI = .97, robust TLI = .95, robust RMSEA = .05 (90% CI: [.03, .08]), and robust SRMR = .06. The likelihood-ratio test was significant, suggesting that our model-implied covariance matrix was not equal to that of the population; that is, our model was not "true" or exact to the population [55]. However, path analyses with large sample sizes provide the smallest confidence intervals around parameter estimates, resulting in the likelihood-ratio test becoming overpowered and more likely to reject models due to small differences in residuals when the test model is well-specified [56].

As a robustness check, we performed the RMSEA test of close fit ($H_0$: model has good fit) and the test of not-close fit ($H_0$: model has poor fit) as outlined by MacCallum and colleagues [57]. For the RMSEA fit index, the asymptotic sampling distribution is known [58, 59], which affords the ability to compute confidence intervals and to conduct null hypothesis significance testing (NHST) about the RMSEA population value. For both the test of close fit and not-close fit, we computed the noncentrality parameter that corresponds to a specified RMSEA cutoff value followed by a one-tailed NHST using the non-central chi-square distribution. For the test of close fit, we specified an RMSEA cutoff value of .05 with NHST results suggesting close model fit as we failed to reject the null hypothesis, $\chi^2(24) = 61.38$, $p > .05$. For the test of not-close fit, we specified an RMSEA cutoff value of .11 with NHST results suggesting that the

**Table 2. Maximum-likelihood robust models of financial stress, restriction condition, and their interaction predicting reactance to COVID-19 public health restrictions.**

| Effect | $\beta$ (SE) | z-value |
|---|---|---|
| *Predictors* | | |
| Reactance → Adherence to Social Distancing | −.42 (.11) | −5.22*** |
| Financial Stress → Reactance | .26 (.10) | 2.88** |
| Restriction Condition → Reactance | .22 (.20) | 1.72[t] |
| Financial Stress*Restriction Condition → Reactance | −.32 (.13) | −2.19* |
| *Covariates* | | |
| PSOC →Reactance | −.27 (.05) | −4.35*** |
| Political Orientation →Reactance | .35 (.04) | 5.24*** |
| Income →Reactance | −.07 (.02) | −1.29 |
| Gender →Reactance | −.04 (.10) | −0.67 |
| *Covariances* | | |
| Restriction Condition ↔ Political Orientation | .02 (.02) | 0.85 |
| Restriction Condition ↔ Interaction | .83 (.01) | 35.39*** |
| Financial Stress ↔ Income | −.23 (.08) | −5.37*** |
| Financial Stress ↔ Interaction | .36 (.05) | 4.82*** |

$N$ = 301. PSOC = Psychological sense of community.

[t] $< .10$,

* $p < .05$,

** $p < .01$,

*** $p < .001$.

model does not have poor fit as we reject the null hypothesis, $\chi^2(24) = 80.08$, $p < .05$. Consequently, these results combined with the other comparative and absolute fit indices indicate that our model has good fit despite the significant likelihood-ratio test. Although restriction condition did not significantly predict reactance, $b = .35$, $p = .09$, 95% CI [-0.05, 0.75], there was a significant main effect of financial stress ($b = .30$, $p = .004$, 95% CI [0.09, 0.50]), and the interaction of financial stress with restriction condition was significant ($b = -.29$, $p =. 03$, 95% CI [-0.55, -0.03]). For the two-way interaction, simple slopes analysis revealed that financial stress was positively associated with state reactance for those in the self-focused condition, $b = 0.29$, SE = .10, $p = .004$. However, for those in the community-focused condition, there was not a significant association between financial stress and reactance, $b = 0.01$, SE = .09, $p = .96$. To contextualize the interaction, we performed a second simple slopes analysis with financial stress moderating restriction condition, allowing us to probe for differences between restriction condition at the mean level of financial stress as well as one standard deviation above and below the mean. Among individuals experiencing high levels of financial stress (+1 $SD$ = 2.21), being in the community-focused condition predicted less reactance than being in the self-focused condition ($b = -0.29$, SE = .13, $p = .03$), but reactance did not differ significantly between conditions at mean ($M = 1.49$, $b = -0.08$, SE = .08 $p = .30$) or low levels of financial stress (-1 $SD$ = .08, $b = 0.12$, SE = .12 $p = .27$; Fig 3).

For covariates, PSOC significantly predicted decreased reactance ($b = -.21$, $p < .001$ 95% C1 [-0.30, -0.11]) and political orientation significantly predicted increased reactance ($b = .19$, $p < .001$, 95% CI [0.12, 0.26]). Recall that a higher score for political orientation represents a more conservative political orientation, suggesting that more conservative individuals had greater reactance. Income ($b = -0.02$, $p = .20$, 95% C1 [-0.06, 0.01]) and gender ($b = -0.07$, $p = .50$, 95% CI [-0.13, 0.26]) did not predict reactance.

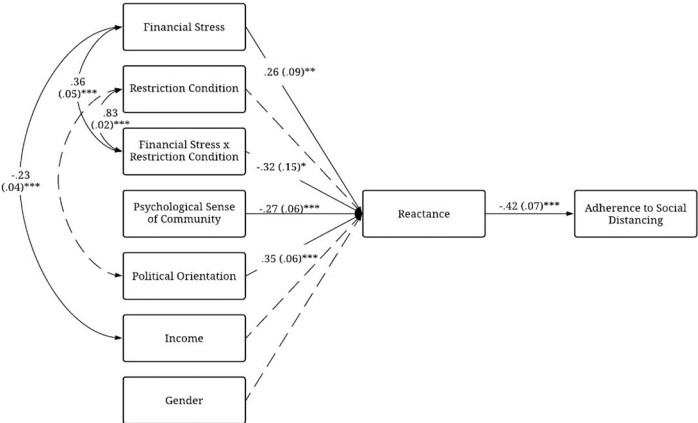

**Fig 2. Path model predicting adherence to social distancing guidelines and reactance to COVID-19 public health restrictions from financial stress, restriction condition (self- or community-focused), and the interaction between financial stress and restriction condition while controlling for gender, psychological sense of community, political orientation, and income.** Values are unstandardized regression estimates with standard errors in parentheses. $^*$ $p <$ .05, $^{**}$ $p < $ .01, $^{***}$ $p < $ .001.

Finally, reactance was significantly associated with adherence to social distancing guidelines, $b$ = -0.57, $p < $ .001, 95% CI [-0.78, -0.35]. That is, those reporting greater reactance also reported lower levels of social distancing behavior. Overall, the variance explained by the model for state reactance was $R^2$ = .25 and for adherence to social distancing was $R^2$ = .18 with an adjusted GFI = .99 for the entire model. This suggests that 25% of the variance in state reactance, reflecting a large effect size, 18% of the variance in adherence to social distancing, reflecting a medium-to-large effect size, and that 99% of the variance in the sample covariance matrix were accounted by the model.

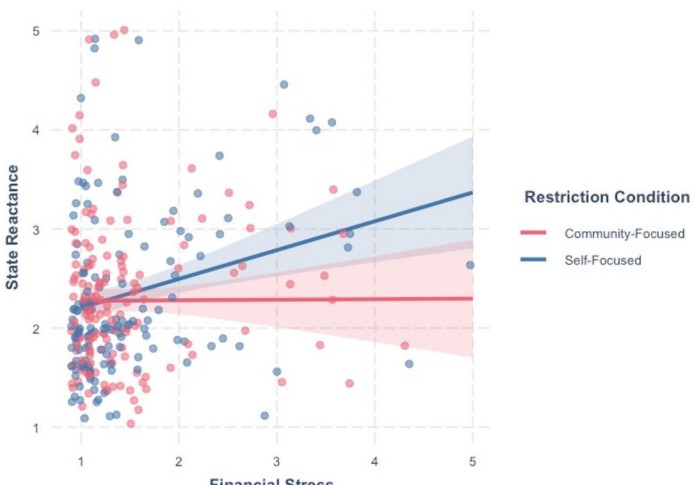

**Fig 3. Simple slopes analyses for the interaction between restriction condition (self- and community-focused) and financial stress predicting state reactance to COVID-19 public health restrictions, controlling for gender, psychological sense of community, political orientation, and income.** Financial stress was positively associated with state reactance for those in the self-focused condition but was nonsignificant in the community-focused condition.

## Exploratory post hoc analysis

Given that, despite random assignment to condition, more politically conservative individuals were over-represented in the community-focused condition, we examined a second, exploratory path model that included a restriction condition x political orientation interaction effect. The exploratory path model had poor fit, $\chi^2(30) = 274.31$, $p < .001$, robust CFI = .74, robust TLI = .61, robust RMSEA = .19 (90% CI: [.17, .21]), and robust SRMR = .13. Moreover, the restriction condition x political orientation interaction effect was not significant, $b = -0.01$, $p = .85$, 95% CI [-0.15, 0.13], and including it in the model did not change the nonsignificant main effect for restriction condition, $b = .39$, $p = .09$, 95% CI [-0.05, 0.83] nor the significant restriction condition x financial stress interaction effect, $b = -0.29$, $p = .03$, 95% CI [-0.56, -0.03].

## Discussion

Soon after governments in the U.S. began issuing public health guidelines and regulations to stem the transmission of COVID-19 in the early spring of 2020, people began complaining, defying, and protesting against these measures [16]. The current study demonstrates that adults who experienced more pandemic-related financial stress were more likely to react against restrictive public health measures. In accordance with the vicarious reactance tenet of psychological reactance theory (PRT), however, public health messaging that drew focus to the effects of such measures on the community, rather than the self, attenuated financially-stressed adults' reactance. These effects were independent of adults' actual income and gender, and over-and-above the tendencies for more conservative individuals to report greater reactance against regulations, and those who felt more connected with COVID-affected groups (PSOC) to report less reactance. Notably, greater reactance towards COVID-19 public health messaging was associated with lower adherence with social distancing guidelines, a critical preventive health behavior in the pandemic.

These findings have implications both for understanding why some individuals have actively resisted cooperating with public health regulations and for developing more effective messaging to solicit their greater cooperation in future. Moreover, the rapid spread and climbing infection rate from the SARS-CoV-2 B.1.617.2 variant (i.e., the delta variant) has instigated worldwide discussions of implementing COVID-19 vaccine mandates [60] and re-instating mask mandates even for those who are vaccinated [61]. Emerging research suggests that vaccine mandates for adults will likely prompt anger, heightened reactance, and defiance and backlash towards such mandates [62, 63]. Our findings build upon current COVID-19 health messaging research by underscoring the need to consider the contextual factors specific to COVID-19 in combination with message framing to better tailor public health messaging, decrease reactance towards mandates, and elicit greater compliance with public health guidelines.

The lack of a main effect of restriction condition suggests that, contrary to previous PRT research [20] and our hypotheses, the intensity of personal and vicarious reactance did not differ significantly. However, our findings are consistent with recent research that has also failed to find differences between self- and other-focused messages in reactance [27] or in COVID-19 attitudes and beliefs [26]. It is possible that the COVID-19 pandemic and associated public health measures had been so pervasive and such drastic deviations from everyday life in the U.S. that the framing of messaging, in and of itself, did not influence people's negative responses to continued freedom loss. Some previous investigations of self-focused and vicarious reactance also failed to find differences between restriction conditions on state reactance [27, 45], or differences that were significant only in the context of moderator variables [21]. Such findings suggest that differences between self-focused and vicarious reactance could depend upon

other factors, such as pandemic-related financial stress as observed in the present study. Additionally, these findings in addition to others, highlight the complexity of responses to the pandemic, such that messages which imply that people have agency in spreading the virus, rather than lower control, may lead people to having less reactive emotions to restrictions [27]. Overall, message framing and the intricacies of the source, target, among other factors are worthy of consideration when attempting to understand responses to public health messaging.

The vignettes asked participants to reflect upon how restrictions would continue to impact the self or community in the *future*, which may have prompted anxiety or anger about limitations and hardships continuing indefinitely for people who, given other studies, likely were already feeling stressed [64]. Clearly, imagining continued employment disruptions elicited considerable negativity and defiance in those currently feeling financial anxiety about making ends meet, yet their reactance was diminished when their attention was focused on how continued public health measures would affect the community. The vicarious reactance condition should have led participants to adopt a more cognitive, reflective approach to considering the situation, rather than an emotional, impulsive stance [5, 21]. Thus, reminders that "we are all in this together" may lessen reactance in financially stressed individuals as they contemplate how others may be equally or more impacted by public health measures, and conversely, how their own situation may not be as uniquely difficult as imagined. Looking toward the future and the possibilities for continuing public health measures in response to COVID-19 variant strains, or other yet-to-be identified pandemics, we propose that community-focused messaging regarding individual and societal behavioral changes may help to curb reactance in those who feel personally adversely affected, which may increase cooperation and the success of public health efforts.

Notably, PSOC and political orientation both significantly predicted reactance, elucidating the importance of individual differences in predicting responses to pandemic conditions. Lower reactance in those with greater PSOC with affected groups reemphasizes the importance of community-focused messaging and community-building as an effective public health strategy [23, 24, 65]. Greater reactance in more politically conservative adults mirrors the increasingly polarized U.S. political landscape [66] which has influenced how people perceive and comply with public health regulations and messages. People are more receptive towards messages that align with their attitudes [67, 68], and more reactant towards messages that challenge their identity or beliefs [69, 70]. Further research is warranted to better understand the role of political orientation in public health messaging and compliance with regulations, but emphasizing the benefits of public health measures for the communities and priorities with which more conservative adults identify may be one way to enlist their cooperation.

## Strengths and limitations

Our study had many strengths. We measured peoples' reactance soon after widespread implementation of COVID-19 emergency orders, revealing how the public acutely responded to an unprecedented global health crisis; we examined responses in a large, geographically diverse sample; and the randomized manipulation of restriction scenarios provided evidence that focusing on one's community produces lower reactance in financially-stressed adults. Our findings expand upon the growing literature exploring the impact of prosocial, other-oriented appeals for public health compliance [23, 37], providing nuance in demonstrating reactance and pandemic-specific stressors as key processes underlying associations between message framing and health behavior. With increasing uncertainty around emerging variants, vaccine booster shots, and a possible return to mask mandates, our findings provide critical insight into the direct and interactive influences of message framing and pandemic-specific stressors

in dampening the "boomerang effect" and promoting health behavior compliance as the pandemic continues.

Of note, our study measured individual's reported reactance to COVID-19 public health restriction scenarios and the extent to which they tried to comply with social distancing; we did not ask participants to report their actual compliance with COVID-19 public health regulations or their perceived risk of COVID-19. Additionally, we focused on social distancing behavior exclusively as mask wearing had not been mandated and was actively cautioned against at the time of study design [71, 72]. Thus, we have a limited understanding of how financial stress, message framing, and reactance uniquely and jointly predict other types of pandemic public health behaviors, such as wearing a mask or receiving the vaccine.

Additionally, we measured a limited number of individual difference variables. Emerging research suggests that personality traits are differentially associated with pandemic response and COVID-19 health behaviors [73]. Following a socioeconomic determinants of health perspective [74], an individual's occupation (e.g., frontline worker, essential worker) may also influence the individual's attitudes and beliefs towards COVID-19, the associated public health mandates, and compliance with preventative health behavior. We did not collect data on personality and occupational status, but future research should explore how these additional individual difference factors relate to public health message framing, reactance, and COVID-19 public health compliance.

Despite random assignment to restriction condition, the disproportionate representation of more politically conservative participants within the community-focused group may have elevated reactance within this condition and reduced the magnitude of the predicted difference between restriction conditions; future research should explore this possibility further. Additionally, there was a disproportionate distribution of gender in our study. The predominance of females in our sample limits the generalizability of our findings to other genders. The majority of research on gender effects in response to COVID-19 has found greater outrage, reactance, and public health noncompliance in males [2, 24, 44], but this was not evident in our study. It behooves future research to further explore how public health message framing influences reactance in individuals who do not identify as female.

Finally, the 2020 U.S. federal election campaign and the nationwide community actions sparked by the murder of George Floyd on May 25, 2020, overlapped with our data collection; either may have affected how participants engaged with questions pertaining to government mandates. Additionally, our sample was predominantly White. Given that COVID-19 has disproportionately impacted Black and Latinx communities [75, 76], concerted efforts are needed to better understand how to best support these communities, including whether different approaches to public health messaging may be needed to more effectively reduce their risks.

## Conclusion

Community-focused messaging reduced reactance to public health regulations restricting activities in adults already feeling financially stressed due to efforts to curb transmission of the COVID-19 pandemic. This suggests that careful tailoring of public health messages may be effective for increasing compliance and cooperation, or at least limiting the harm of pushing against regulations. Ultimately, the success of the nation in appealing to citizens regarding preventative health behavior and restrictions is partly due to the way regulations are tailored. The economic, social, and personal effects of the pandemic have been tremendous, and at times overwhelming, for many people, yet shifting the frame of their focus towards the community may limit some of the emotional impact of ongoing turmoil due to the pandemic.

## Acknowledgments

We thank all participants for their time and participation in this study and Heather Elahi for her assistance with preparing the document.

## Author Contributions

**Conceptualization:** Michael E. Knapp, Lindsey C. Partington, Ryan T. Hodge, Elisa Ugarte, Paul D. Hastings.

**Data curation:** Lindsey C. Partington, Ryan T. Hodge, Elisa Ugarte.

**Formal analysis:** Michael E. Knapp, Lindsey C. Partington, Paul D. Hastings.

**Investigation:** Michael E. Knapp, Lindsey C. Partington, Ryan T. Hodge, Elisa Ugarte, Paul D. Hastings.

**Methodology:** Michael E. Knapp, Lindsey C. Partington, Paul D. Hastings.

**Project administration:** Paul D. Hastings.

**Supervision:** Paul D. Hastings.

**Visualization:** Michael E. Knapp, Lindsey C. Partington.

**Writing – original draft:** Michael E. Knapp, Lindsey C. Partington.

**Writing – review & editing:** Michael E. Knapp, Lindsey C. Partington, Ryan T. Hodge, Elisa Ugarte, Paul D. Hastings.

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
