## [Decision Letter · Decision Letter 0]

21 Jul 2021

PONE-D-21-19560

We’re all in this together: Focus on community attenuates effects of pandemic-related financial hardship on reactance to COVID-19 public health regulations

PLOS ONE

Dear Dr. Knapp,

Thank you for submitting your manuscript to PLOS ONE. After careful consideration, we feel that it has merit but does not fully meet PLOS ONE’s publication criteria as it currently stands. Therefore, we invite you to submit a revised version of the manuscript that addresses the points raised during the review process.

Please find below the reviewer's comments, as well as those of mine.

We look forward to receiving your revised manuscript.

Kind regards,

Valerio Capraro

Academic Editor

PLOS ONE

Journal Requirements:

2. Please ensure that you include a title page within your main document. We do appreciate that you have a title page document uploaded as a separate file, however, as per our author guidelines (http://journals.plos.org/plosone/s/submission-guidelines#loc-title-page) we do require this to be part of the manuscript file itself and not uploaded separately. Could you therefore please include the title page into the beginning of your manuscript file itself, listing all authors and affiliations.

Additional Editor Comments (if provided):

I have now collected one review from one expert in the field. I was unable to find a second reviewer, but I am myself familiar with the topic of this manuscript, so I feel confident in making a decision with only one review. As you will see, the review is positive, but suggests a major revision. I tend to agree with the reviewer, therefore I would like to invite you to revise your manuscript for Plos One. On top of the reviewer's comments, I would like to add two more comments, mainly related to a potential effect of gender, and to the literature review. Indeed, I have noticed that in your sample there are many more females than males. Since there are gender differences in pandemic response (Capraro & Barcelo, 2020; Haischer et al. 2020), it is important to control for gender (it seems to me that you did not do so). Regarding the literature review, I was surprised to see that you ignored the previous papers on the effect of prosocial, community-oriented, messages on pandemic response, which seem to be directly related to your work (Capraro & Barcelo, 2020; Jordan et al., 2020). More generally, there has been a number of papers looking at the effect of prosocial messages on pandemic response (see Capraro et al. 2021 for a review). Another work that you might find useful is the perspective article on what social and behavioral science can do to support pandemic response published by Van Bavel et al. in Nature Human Behaviour.

I am looking forward for the revision.

Capraro, V., & Barcelo, H. (2020). The effect of messaging and gender on intentions to wear a face covering to slow down COVID-19 transmission. Journal of Behavioral Economics for Policy, 4, Special Issue 2, 45-55.

Capraro, V., Boggio, P.S., Böhm, R., Perc, M., & Sjåstad, H. (forthcoming) Cooperation and acting for the greater good during the COVID-19 pandemic. In M. K. Miller (Ed.) The

Social Science of the COVID-19 Pandemic: A Call to Action for Researchers. Oxford: Oxford University Press. Available at: https://doi.org/10.31234/osf.io/65xmg

Haischer, M. H., Beilfuss, R., Hart, M. R., Opielinski, L., Wrucke, D., Zirgaitis, G., ... & Hunter, S. K. (2020). Who is wearing a mask? Gender-, age-, and location-related differences during the COVID-19 pandemic. Plos one, 15(10), e0240785.

Jordan, J., Yoeli, E., & Rand, D. (2020). Don’t get it or don’t spread it? Comparing self-interested versus prosocially framed COVID-19 prevention messaging. https://psyarxiv.com/yuq7x

Van Bavel, J. J., et al. (2020). Using social and behavioural science to support COVID-19 pandemic response. Nature Human Behaviour, 4, 460-471.

Reviewers' comments:

Reviewer's Responses to Questions

**Comments to the Author**

1. Is the manuscript technically sound, and do the data support the conclusions?

Reviewer #1: Partly

2. Has the statistical analysis been performed appropriately and rigorously? 

Reviewer #1: Yes

3. Have the authors made all data underlying the findings in their manuscript fully available?

Reviewer #1: No

4. Is the manuscript presented in an intelligible fashion and written in standard English?

Reviewer #1: Yes

5. Review Comments to the Author

Reviewer #1: Thank you very much for the opportunity to revise this manuscript. I believe the manuscript has the potential to add to existing research on compliance with COVID-19 Public Health Regulations. Below, I state my comments that I hope you would find it useful.

1- I appreciate the authors’ introduction in terms of the contribution of the psychological reactance theory to this literature. However, all of a sudden, the manuscript started to present thoughts about republicans, and democrats in page 5. There is a lack of flow in the text in this part. The political orientation in fact seems very disconnected from the basic premises of the work.

2- The way I read the empirical arguments is that Finanical stress results in reactance towards COVID-19 public health restrictions, which in turn affect social distancing behavior. Individual differences in PSOC, and political orientation moderate these associations. Yet, the hypotheses do not explain at which step do the moderators take place. Neither does the model in Figure 2.

3- The replicability link does not work, showing that the depository is empty.

4- Further descriptives are needed, such as the distribution of the main variables.

5- I wonder what is the reason for using path analysis for the direct effects? A general OLS for the direct effects and give a better glimpse about the effect sizes.

6- I wonder if the authors could address causality issue in a way or another, specifically, could it be that adherence to social distancing are the one that resulted in financial stress, specially that social distancing rules are the one that could be considered an exogenous shock in this case.

7- Further controls are needed, such as personality traits (i.e. research has shown that stable personality traits such as the big five, and optimism as well as occupations might play a role in compliance with COVD-19 health regulations.

6. PLOS authors have the option to publish the peer review history of their article (what does this mean?). If published, this will include your full peer review and any attached files.

Reviewer #1: No

---

## [Author Response · Author response to Decision Letter 0]

30 Aug 2021

We thank the anonymous Editor and Reviewer for the thorough and helpful reviews they provided to the original manuscript. We have endeavored to respond to each of the comments raised in the revised manuscript and in this response letter. The specific ways in which we have responded to each of the comments is detailed below with our responses provided in italicized font.

Additional Editor Comments (if provided):

I have now collected one review from one expert in the field. I was unable to find a second reviewer, but I am myself familiar with the topic of this manuscript, so I feel confident in making a decision with only one review. As you will see, the review is positive, but suggests a major revision. I tend to agree with the reviewer, therefore I would like to invite you to revise your manuscript for Plos One. On top of the reviewer's comments, I would like to add two more comments, mainly related to a potential effect of gender, and to the literature review.

Indeed, I have noticed that in your sample there are many more females than males. Since there are gender differences in pandemic response (Capraro & Barcelo, 2020; Haischer et al. 2020), it is important to control for gender (it seems to me that you did not do so). 

We originally excluded gender as a covariate as preliminary analyses indicated that it was not significantly correlated with any of the key study variables nor were there any mean-level differences between gender for any of the key study variables. However, as you have noted, the COVID-19 literature has found notable gender differences in pandemic response. In the introduction, we have included a paragraph describing gender differences in pandemic response and how it may relate to reactance (p. 5-6). Additionally, we have re-run our path analysis to include gender as a covariate and have updated the Results and associated Tables and Figures to reflect the new model. Additionally, we are heartened that key findings were maintained in the new model controlling for gender. 

Regarding the literature review, I was surprised to see that you ignored the previous papers on the effect of prosocial, community-oriented, messages on pandemic response, which seem to be directly related to your work (Capraro & Barcelo, 2020; Jordan et al., 2020). More generally, there has been a number of papers looking at the effect of prosocial messages on pandemic response (see Capraro et al. 2021 for a review). Another work that you might find useful is the perspective article on what social and behavioral science can do to support pandemic response published by Van Bavel et al. in Nature Human Behaviour.

I am looking forward for the revision.

Capraro, V., & Barcelo, H. (2020). The effect of messaging and gender on intentions to wear a face covering to slow down COVID-19 transmission. Journal of Behavioral Economics for Policy, 4, Special Issue 2, 45-55.

Capraro, V., Boggio, P.S., Böhm, R., Perc, M., & Sjåstad, H. (forthcoming) Cooperation and acting for the greater good during the COVID-19 pandemic. In M. K. Miller (Ed.) The Social Science of the COVID-19 Pandemic: A Call to Action for Researchers. Oxford: Oxford University Press. https://doi.org/10.31234/osf.io/65xmg

Haischer, M. H., Beilfuss, R., Hart, M. R., Opielinski, L., Wrucke, D., Zirgaitis, G., ... & Hunter, S. K. (2020). Who is wearing a mask? Gender-, age-, and location-related differences during the COVID-19 pandemic. Plos one, 15(10), e0240785.

Jordan, J., Yoeli, E., & Rand, D. (2020). Don’t get it or don’t spread it? Comparing self-interested versus prosocially framed COVID-19 prevention messaging. https://psyarxiv.com/yuq7x

Van Bavel, J. J., et al. (2020). Using social and behavioural science to support COVID-19 pandemic response. Nature Human Behaviour, 4, 460-471.

We appreciate the Editor’s recommended literature and have incorporated the suggested papers into our introduction. Structurally, we have added the following subheadings to the introduction: Reactance in the Pandemic Context (p. 2-4), Avenues for Attenuating Reactance: Other-Oriented Messaging (p. 4-5), Pandemic-Specific Financial Stress as a Moderating Influence (p. 5-6), and Critical Covariates: Individual Differences in Pandemic Response (p. 6-8). We have incorporated your suggested literature into the Avenues for Attenuating Reactance: Other-Oriented Messaging subsection (p. 4-5) where we have expanded upon public health message framing and differential impacts on COVID-19 preventative behavior and public health compliance. Additionally, we have expanded our Discussion to include how our findings relate back to the growing literature regarding prosocial, other-oriented messages in the context of COVID-19 (p. 21-22, 24-25). 

Reviewers' comments:

Reviewer's Responses to Questions

Comments to the Author

1. Is the manuscript technically sound, and do the data support the conclusions?

Reviewer #1: Partly

2. Has the statistical analysis been performed appropriately and rigorously?

Reviewer #1: Yes

3. Have the authors made all data underlying the findings in their manuscript fully available?

Reviewer #1: No

4. Is the manuscript presented in an intelligible fashion and written in standard English?

Reviewer #1: Yes

5. Review Comments to the Author

Reviewer #1: Thank you very much for the opportunity to revise this manuscript. I believe the manuscript has the potential to add to existing research on compliance with COVID-19 Public Health Regulations. Below, I state my comments that I hope you would find it useful.

1- I appreciate the authors’ introduction in terms of the contribution of the psychological reactance theory to this literature. However, all of a sudden, the manuscript started to present thoughts about republicans, and democrats in page 5. There is a lack of flow in the text in this part. The political orientation in fact seems very disconnected from the basic premises of the work.

We appreciate the Reviewer’s comments regarding manuscript flow and cohesion. We have restructured the introduction and have added the following subheadings to the introduction: Reactance in the Pandemic Context (p. 2-4), Avenues for Attenuating Reactance: Other-Oriented Messaging (p. 4-5), Pandemic-Specific Financial Stress as a Moderating Influence (p. 5-6), and Critical Covariates: Individual Differences in Pandemic Response (p. 6-8). We have expanded upon political orientation as a salient individual difference factor in the context of COVID-19, providing further explanation as to how political orientation may relate to reactance and therefore should be controlled for in subsequent analyses (p. 7-8). 

2- The way I read the empirical arguments is that Finanical stress results in reactance towards COVID-19 public health restrictions, which in turn affect social distancing behavior. Individual differences in PSOC, and political orientation moderate these associations. Yet, the hypotheses do not explain at which step do the moderators take place. Neither does the model in Figure 2.

We appreciate the Reviewer’s statement of their interpretation of the empirical arguments; however, it was not our intent to suggest that PSOC and political orientation moderate these associations. Rather, our argument is that PSOC and political orientation may be associated with how people respond to COVID-19 public health messaging; thus, they are necessary covariates for analyses aimed at examining the direct and interactive effects of restriction condition and pandemic-specific financial stress. We have re-structured and expanded the introduction to better articulate our empirical arguments and hypotheses. Additionally, we have re-ordered the Measures subsection to present focal measures related to hypotheses first, followed by covariates (p. 11-14). Hopefully, our line of reasoning is conveyed more clearly now. 

Additionally, to ensure that we were not erring in treating these measures as covariates rather than moderators, we tested alternative models that included moderating effects of political orientation and PSOC on restriction condition and reactance. Neither interaction term was significant: political orientation x restriction condition (b = -.02, p = .73); PSOC x restriction condition (b = .04, p = .67). 

3- The replicability link does not work, showing that the depository is empty.

We apologize for the oversight. It was our understanding that the replicability link could be provided upon acceptance of the manuscript; therefore, it was not provided. We have corrected our error and the replicability link should now work.

4- Further descriptives are needed, such as the distribution of the main variables.

We have added information on skew and kurtosis to Table 1 (p. 12-13) for all key study variables to provide further information about the distribution of the main variables. If other descriptive statistics are needed then we are happy to provide them. 

5- I wonder what is the reason for using path analysis for the direct effects? A general OLS for the direct effects and give a better glimpse about the effect sizes.

Although OLS regression may be considered a more straightforward or accessible approach to addressing the direct effects, we opted to use path models in a structural equation modelling framework as this provides numerous quantitative advantages over OLS. To be succinct, SEM affords more model specification flexibility, generates fit indices, can correct for multivariate nonnormality in comparison to OLS, and better handles missing data (Kline, 2011).

To expand on these points, SEM is more flexible and amenable to model specification of hypothesized or known associations between variables whereas OLS does not allow this type of model specification (Kline, 2011). For example, path analysis permits us to model specific covariances between predictors and covariates whereas multiple regression assumes that all covariates and predictors meaningfully covary. SEM also allows us to model predictive paths between multiple endogenous variables in one model, affording a more parsimonious analysis in comparison to multiple OLS models. SEM allows us to impose a specific covariance structure that reflects theory or hypothesized associations between variables, allowing us to then test “goodness-of-fit” of this theoretically-based covariance structure (Guttman, 1971; Hayduk et al., 2007). Additionally, SEM provides us with multiple model fit indices to assess how well our proposed model reflects the covariance in the data and the extent to which our hypothesized associations are plausible in the population whereas OLS only provides effect size of variance explained by the model, but not fit (Kline, 2011; MacCallum & Austin, 2000). OLS assumes multivariate normality, an assumption which our data has violated. Unfortunately, OLS does not afford robust corrections to violations of multivariate normality whereas SEM can correct for this using Maximum Likelihood with Robust Corrections estimation (Mardia, 1971; Satorra & Bentler, 1994). Finally, SEM allowed us to use sophisticated techniques like Full Information Maximum Likelihood for estimating missing data (Kline, 2011). 

Yet, as a robustness check, we have examined two hierarchical multiple regression models to test our hypotheses predicting reactance and one linear multiple regression model to test our hypothesis for reactance predicting adherence to social distancing behavior. Please note that multivariate normality was violated in these analyses. The results of the two hierarchical models indicate that the model with the interaction term explains significantly more variance than the model without the interaction term. Moreover, all of our findings from the SEM are maintained in the model with the interaction term. In the final model predicting adherence to social distancing guidelines, we found continued support for our SEM findings with greater reactance significantly predicting decreased adherence to social distancing. The regression models did not identify new associations that were not evident in SEM. We are heartened by the robustness of our findings across analytic approaches, and we continue to utilize and report the SEM analyses because of their considerable quantitative advantages. 

We understand the Reviewer’s concern regarding OLS and effect size. We have reported both R2 and GFI but have also added the following verbiage to increase clarity: 

“This suggests that 25% of the variance in state reactance, reflecting a large effect size, 18% of the variance in adherence to social distancing, reflecting a medium-to-large effect size, and that 99% of the variance in the sample covariance matrix were accounted by the model.” (p. 20)

6- I wonder if the authors could address causality issue in a way or another, specifically, could it be that adherence to social distancing are the one that resulted in financial stress, specially that social distancing rules are the one that could be considered an exogenous shock in this case.

We understand the Reviewer’s concern regarding alternative models and causality and appreciate the Reviewer’s suggestion of examining the effects of social distancing as an exogeneous shock. Of course, the experimental component of this study was the random assignment of participants to the self-focused versus community-focused restriction condition, and causal inferences should only be drawn about the borderline-significant main effect of restriction condition. Individual variation in financial stress may be seen as a result of the “natural experiment” of COVID-19, but lacking random assignment or pre-COVID measures, caution must be exercised regarding the inference of causal or directional relations between financial stress and reactance against (or adherence with) public health restrictions.

That said, we tested the model that the Reviewer suggested, removing the path in which reactance predicts adherence to social distancing and adding a path in which adherence to social distancing predicts financial stress. The alternative model has good to adequate fit, χ^2 = 57.49, p < .001, robust CFI = .94, robust TLI = .90, robust RMSEA = .08 (90% CI: [.06, .11]), robust SRMR = .07. However, these fit indices are inferior to those in our theorized model. Additionally, the alternative model has a higher AIC and BIC value (AIC = 6537.34, sample-size adjusted BIC = 6555.02) in comparison to our theorized model (AIC = 6515.71, sample-size adjusted BIC = 6531.78). Finally, we calculated an AIC evidence ratio (Burnham et al., 2011) of 49786.54, suggesting that the theorized model is approximately 50,000 times stronger than the alternative model. Of note, adherence to social distancing significantly predicted decreased financial stress (b = -0.08, p = .02). From a theoretical perspective, we would expect this path coefficient to be positive wherein social distancing is contributing to increased financial stress, perhaps by impacting ability to work; however, the quantitative evidence does not suggest this. We are encouraged by these results and will continue to report and interpret the findings from our theorized model. We have not reported the alternate model in the revised manuscript; however, if the Reviewer and Editor think that it is important information to include, we would be happy to include the alternate model in the main text or an online supplement.

We would also like to emphasize that our measure of financial stress was designed to capture a wide-range of economic impacts from the pandemic, some of which would not be theoretically linked to social distancing. We have added more example items from the measure to increase clarity for the reader (p. 12). 

7- Further controls are needed, such as personality traits (i.e. research has shown that stable personality traits such as the big five, and optimism as well as occupations might play a role in compliance with COVD-19 health regulations.

We appreciate the Reviewer’s suggestion of controlling for personality traits and work occupation; unfortunately, these variables were not measured in our study and cannot be included as covariates. However, we have included gender as an additional covariate with no impact to our findings. In the Discussion, we have also recommended examination of personality traits and occupational status as avenues for future research. 

“Additionally, we measured a limited number of individual difference variables. Emerging research suggests that personality traits are differentially associated with pandemic response and COVID-19 health behaviors (Corr, 2021). Following a socioeconomic determinants of health perspective (Khalatbari-Soltani et al., 2020), an individual’s occupation (e.g., frontline worker, essential worker) may also influence the individual’s attitudes and beliefs towards COVID-19, the associated public health mandates, and compliance with preventative health behavior. We did not collect data on personality and occupational status, but future research should explore how these additional individual difference factors relate to public health message framing, reactance, and COVID-19 public health compliance.” (p. 24-25)

---

## [Decision Letter · Decision Letter 1]

12 Oct 2021

PONE-D-21-19560R1We’re all in this together: Focus on community attenuates effects of pandemic-related financial hardship on reactance to COVID-19 public health regulationsPLOS ONE

Dear Dr. Knapp,

Thank you for submitting your manuscript to PLOS ONE. After careful consideration, we feel that it has merit but does not fully meet PLOS ONE’s publication criteria as it currently stands. Therefore, we invite you to submit a revised version of the manuscript that addresses the points raised during the review process. I was able to secure one review from an expert in the field. The reviewer has provided excellent feedback and has highlighted some remaining concerns. Please be sure to address all of these remaining issues in your revision.Of particular importance, the reviewer noted that the replicability link still does not work. 

We look forward to receiving your revised manuscript.

Kind regards,

Neha John-Henderson

Academic Editor

PLOS ONE

Journal Requirements:

Reviewers' comments:

Reviewer's Responses to Questions

**Comments to the Author**

1. If the authors have adequately addressed your comments raised in a previous round of review and you feel that this manuscript is now acceptable for publication, you may indicate that here to bypass the “Comments to the Author” section, enter your conflict of interest statement in the “Confidential to Editor” section, and submit your "Accept" recommendation.

Reviewer #2: (No Response)

2. Is the manuscript technically sound, and do the data support the conclusions?

Reviewer #2: Partly

3. Has the statistical analysis been performed appropriately and rigorously? 

Reviewer #2: Yes

4. Have the authors made all data underlying the findings in their manuscript fully available?

Reviewer #2: No

5. Is the manuscript presented in an intelligible fashion and written in standard English?

Reviewer #2: Yes

6. Review Comments to the Author

Reviewer #2: The authors sufficiently responded to the comments from the Editor and Reviewer. I appreciate the amount of detail and effort that went into each of the responses. In particular, I appreciate the inclusion of gender in the analyses. I also appreciate the inclusion of subheadings in the introduction, as this makes for an easier read.

MAJOR NOTES

-Following up on the original comment from Reviewer 1, the replicability link still does not work, showing that the depository is empty.

-Where did the item for “Adherence to Social Distancing” come from? Is this pulled from another questionnaire, or was it developed by the authors themselves for the purpose of this study? Please state either way.

-Was the wording of the 6 items from Conger and colleagues’ economic hardship measures adapted to include “since the pandemic began”? If so, please state that you adapted/changed this measure to meet the needs of the current study.

-Under “Psychological Sense of Community”, you provide an example item for the two items measuring personal identification, but you do not provide an example item for the two items measuring connection. While not a huge deal, I recommend also including an example item of connection for your readers.

-You don’t mention income as a covariate in your introduction, but you mention it in your method section, and even use it as an exclusion criterion if not answered. Perhaps briefly mention it in your “Current Study” section of the introduction.

-Table 1 is very helpful, and I appreciate its inclusion in the manuscript. However, it does not follow APA formatting guidelines. Additionally, many of the values do not line up, such that some are flush left, and some are centered, making it harder to read. Please reformat this table (e.g., remove all vertical lines and any excess horizontal lines, center all values, etc.).

-You mention that those in the community-focused condition were more politically conservative (M = 2.99, SD = 1.46) compared to those in the self-focused condition, M = 2.57, SD = 1.53, t(299) = -2.44, p = .015, 95% CI: [-0.76, -0.08]. Do you think this could have impacted the lack of effect that restriction condition had on reactance, given that political orientation significantly impacted reactance? It is possible I missed where this was discussed or ruled out in the analyses. But if not, please make mention of this either in your analyses or briefly in your discussion.

-Why is gender listed in Fig. 3 but not Fig. 2?

-The disproportionate distribution of gender was not mentioned in your limitations section. I recommend mentioning this alongside the statement about ethnicity.

MINOR NOTES

- Throughout the introduction, the United States is sometimes written as U.S. and other times is written as US. Please keep consistent throughout.

-For figures 1a and 1b: in the text, the a and b are lowercase but, in the figures, they are uppercase (A; B). Please keep consistent.

-In some places, you round to three decimals, and in other places, you round to two. For example, the Cronbach’s alpha for State Reactance was .896, 95% CI [.867, .917]. I recommend keeping it to two decimals throughout.

-Under the “State Reactance” subheading, please change the following sentence from “The scale consisted of 11items assessing experience of reactance” to “The scale consisted of 11 items assessing experience of reactance” (insert space between 11 and items).

-You included gender in your Fig. 3 model, but it is not named in the caption for Fig. 3. Please include alongside mention of the other variables.

7. PLOS authors have the option to publish the peer review history of their article (what does this mean?). If published, this will include your full peer review and any attached files.

Reviewer #2: No

---

## [Author Response · Author response to Decision Letter 1]

10 Nov 2021

We thank the anonymous Reviewer for the thorough and helpful reviews they provided to the original manuscript. We have endeavored to respond to each of the comments raised in the revised manuscript and in this response letter. The specific ways in which we have responded to each of the comments is detailed below with our responses provided in italicized font.

Reviewers' comments:

6. Review Comments to the Author

Reviewer #2: The authors sufficiently responded to the comments from the Editor and Reviewer. I appreciate the amount of detail and effort that went into each of the responses. In particular, I appreciate the inclusion of gender in the analyses. I also appreciate the inclusion of subheadings in the introduction, as this makes for an easier read. MAJOR NOTES

 Following up on the original comment from Reviewer 1, the replicability link still does not work, showing that the depository is empty. 

We are able to view and download the data and associated analytic script when we click the replicability link, and we are unsure why the depository is empty for both you and the Editor. Since receiving the revision request, we have deleted and reuploaded the files, and had multiple lab members check and they were able to view and download the files. This should be remedied and allow for a complete review.

 Where did the item for “Adherence to Social Distancing” come from? Is this pulled from another questionnaire, or was it developed by the authors themselves for the purpose of this study? Please state either way.

We created this item for the purposes of the study and have clarified this in the “Adherence to Social Distancing” subsection (p. 12). 

 Was the wording of the 6 items from Conger and colleagues’ economic hardship measures adapted to include “since the pandemic began”? If so, please state that you adapted/changed this measure to meet the needs of the current study.

We have added the following sentence to specify that the wording was changed for these measures in order to better capture financial stress in association with COVID-19, “The prompt “Since the pandemic began” was added to items to orient participants to rating financial stress experienced in the wake of the pandemic” (p. 12).

 Under “Psychological Sense of Community”, you provide an example item for the two items measuring personal identification, but you do not provide an example item for the two items measuring connection. While not a huge deal, I recommend also including an example item of connection for your readers.

We appreciate the recommendation and have added the following example item of connection: “Two items measured participants’ connection to those impacted by COVID-19 (e.g., “I feel strong ties to the community affected by COVID-19”) and two additional items measured participants’ personal identification with those impacted by COVID-19 (e.g., “I identify with the community affected by COVID-19”)” (p. 13-14). 

 You don’t mention income as a covariate in your introduction, but you mention it in your method section, and even use it as an exclusion criterion if not answered. Perhaps briefly mention it in your “Current Study” section of the introduction.

Thank you for this suggestion; we now identify pre-pandemic income as a covariate in the Current Study section (p. 9). 

 Table 1 is very helpful, and I appreciate its inclusion in the manuscript. However, it does not follow APA formatting guidelines. Additionally, many of the values do not line up, such that some are flush left, and some are centered, making it harder to read. Please reformat this table (e.g., remove all vertical lines and any excess horizontal lines, center all values, etc.).

Table 1 has been reformatted to meet APA 7 and PLOS One guidelines for Tables. All values are centered, except the first column containing variables names which left-centered for ease of reading associated with long text entries (e.g., names, etc.). Additionally, all unnecessary horizontal and vertical lines have been removed.

 You mention that those in the community-focused condition were more politically conservative (M = 2.99, SD = 1.46) compared to those in the self-focused condition, M = 2.57, SD = 1.53, t(299) = -2.44, p = .015, 95% CI: [-0.76, -0.08]. Do you think this could have impacted the lack of effect that restriction condition had on reactance, given that political orientation significantly impacted reactance? It is possible I missed where this was discussed or ruled out in the analyses. But if not, please make mention of this either in your analyses or briefly in your discussion.

We understand the Reviewer’s concerns and have added an Exploratory Post Hoc Analysis subsection to the Results (pp. 20-21) to address the possibility of a restriction condition x political orientation interaction impacting results (p. 20-21). Of note, including this interaction resulted in poor model fit, χ^2(30) = 274.31, p < .001, robust CFI = .74, robust TLI =.61, robust RMSEA = .19 (90% CI: [.17, .21), and the restriction condition x political orientation interaction was nonsignificant, b = -0.01, p = .85. Moreover, the main effect of restriction condition remained nonsignificant, b = .39, p = .09, while the significant restriction condition x financial stress interaction remained significant, b = -0.29, p = .03. Finally, we have added the following sentence to the Limitations section of the Discussion to acknowledge the possibility of a more politically conservative community-focused group may have influenced the restriction condition main effect: “Despite random assignment to restriction condition, the disproportionate representation of more politically conservative participants within the community-focused group may have elevated reactance within this condition and reduced the magnitude of the predicted difference between restriction conditions; future research should explore this possibility further” (p. 26). Our hope is that these additions to the manuscript better address the potential impact of political orientation on our results. 

 Why is gender listed in Fig. 3 but not Fig. 2?

 The disproportionate distribution of gender was not mentioned in your limitations section. I recommend mentioning this alongside the statement about ethnicity.

We have added a paragraph to the Strengths and Limitations subsection that discusses the disproportionate distribution of gender as a limitation and that contextualizes our findings within the gender and COVID-19 literature (p. 26). 

MINOR NOTES

1. Throughout the introduction, the United States is sometimes written as U.S. and other times is written as US. Please keep consistent throughout.

We have corrected the error and the United States should now be written as U.S. throughout the manuscript. 

2. For figures 1a and 1b: in the text, the a and b are lowercase but, in the figures, they are uppercase (A; B). Please keep consistent.

We have corrected the error and have capitalized A and B within the text to maintain consistency with the figure. 

3. In some places, you round to three decimals, and in other places, you round to two. For example, the Cronbach’s alpha for State Reactance was .896, 95% CI [.867, .917]. I recommend keeping it to two decimals throughout.

We have corrected the error and have all decimals rounded to two decimal places, except for p-values.

4. Under the “State Reactance” subheading, please change the following sentence from “The scale consisted of 11items assessing experience of reactance” to “The scale consisted of 11 items assessing experience of reactance” (insert space between 11 and items).

We have corrected the typological error and have inserted a space between “11” and “items” for this sentence. 

5. You included gender in your Fig. 3 model, but it is not named in the caption for Fig. 3. Please include alongside mention of the other variables.

We have corrected our error and have listed gender in the figure caption.

---

## [Editor Report · Decision Letter 2]

17 Nov 2021

We’re all in this together: Focus on community attenuates effects of pandemic-related financial hardship on reactance to COVID-19 public health regulations

PONE-D-21-19560R2

Dear Dr. Knapp,

We’re pleased to inform you that your manuscript has been judged scientifically suitable for publication and will be formally accepted for publication once it meets all outstanding technical requirements.

Kind regards,

Neha John-Henderson

Academic Editor

PLOS ONE

---

## [Editor Report · Acceptance letter]

15 Dec 2021

PONE-D-21-19560R2 

We’re all in this together: Focus on community attenuates effects of pandemic-related financial hardship on reactance to COVID-19 public health regulations 

Dear Dr. Knapp:

I'm pleased to inform you that your manuscript has been deemed suitable for publication in PLOS ONE. Congratulations! Your manuscript is now with our production department. 

Kind regards, 

on behalf of

Dr. Neha John-Henderson 

Academic Editor

PLOS ONE